Transcriptome analysis reveals the mechanism of pyroptosis-related genes in septic cardiomyopathy

Zhu Haoyan
Wu Jiahe
Li Chenze
Zeng Ziyue
He Tianwen
Liu Xin
Wang Qiongxin
Hu Xiaorong
Lu Zhibing luzhibing222@163.com
Cai Huanhuan caihuanhuan@whu.edu.cn
1 Department of Cardiology, Zhongnan Hospital of Wuhan University , Wuhan , China
2 Institute of Myocardial Injury and Repair, Wuhan University , Wuhan , China
Huang Zunnan
Electronic publication date: 2023 Oct 19
Publication date: 2023
Volume: 11
Electronic Location ID: e16214
Received 2023 Apr 27; Accepted 2023 Sep 11
Copyright: ©2023 Zhu et al.
Copyright year: 2023
Copyright holder: Zhu et al.
License: This is an open access article distributed under the terms of the Creative Commons Attribution License, which permits unrestricted use, distribution, reproduction and adaptation in any medium and for any purpose provided that it is properly attributed. For attribution, the original author(s), title, publication source (PeerJ) and either DOI or URL of the article must be cited.
License URL: https://creativecommons.org/licenses/by/4.0/

Keywords: Bioinformatics analysis, Septic Cardiomyopathy, Pyroptosis, Biomarker, Immune Infiltration

Funding: National Natural Science Foundation of China 82070425 This study was supported by a grant from the National Natural Science Foundation of China (No. 82070425, Zhibing Lu). The funders had no role in study design, data collection and analysis, decision to publish, or preparation of the manuscript.

==============================
Background

Septic cardiomyopathy (SC) is characterized by myocardial dysfunction caused by sepsis and constitutes one of the serious complications of sepsis. Pyroptosis is a unique proinflammatory programmed cell death process. However, the role of pyroptosis in the development of SC remains unclear, and further study is required. The purpose of this study is to identify pyroptosis-related genes (PRGs) in SC and explore the mechanism of pyroptosis involved in the regulation of SC formation and progression.

Methods

Differential expression analysis and enrichment analysis were performed on the SC-related dataset GSE79962 to identify differentially expressed genes (DEGs). PRGs were screened by intersecting genes associated with pyroptosis in previous studies with the DEGs obtained from GSE79962. The expression pattern of them was studied based on their raw expression data. Additionally, corresponding online databases were used to predict miRNAs, transcription factors (TFs) and therapeutic agents of PRGs. Lipopolysaccharide (LPS)-induced cell damage models in H9C2 and AC16 cell lines were constructed, cell activity was detected by CCK-8 and cell pyroptosis were detected by Hoechst33342/PI staining. Furthermore, these PRGs were verified in the external datasets (GSE53007 and GSE142615) and LPS-induced cell damage model. Finally, the effect of siRNA-mediated PRGs knockdown on the pyroptosis phenotype was examined.

Results

A total of 1,206 DEGs were screened, consisting of 663 high-expressed genes and 543 low-expressed genes. Among them, ten PRGs (SOD2, GJA1, TIMP3, TAP1, TIMP1, NOD1, TP53, CPTP, CASP1 and SAT1) were identified, and they were mainly enriched in “Pyroptosis”, “Ferroptosis”, “Longevity regulating pathway”, and “NOD-like receptor signaling pathway”. A total of 147 miRNAs, 31 TFs and 13 therapeutic drugs were predicted targeting the PRGs. The expression trends of SOD2 were confirmed in both the external datasets and LPS-induced cell damage models. Knockdown of SOD2 induced increased pyroptosis in the AC16 LPS-induced cell damage model.

Conclusions

In this study, we demonstrated that SOD2 is highly expressed in both the SC and LPS-induced cell damage models. Knockdown of SOD2 led to a significant increase in pyroptosis in the AC16 LPS-induced cell damage model. These findings suggest that SOD2 may serve as a potential target for the diagnosis and treatment of SC.

Introduction

Septic cardiomyopathy (SC) is a severe myocardial dysfunction caused by sepsis, making it one of the most critical complications of sepsis (Beesley et al., 2018). SC can further lead to severe arrhythmia, heart failure, and sudden cardiac death. In the United States, sepsis carries a high morbidity and mortality rate, imposing a significant economic burden (Fleischmann et al., 2016). Patients with sepsis combined with SC face an even more grim prognosis, with a mortality rate that increases two to three times, reaching up to 70–90% (Lin et al., 2020). SC has a diverse range of etiologies, and currently, there are no specific drugs or treatment options available for this condition. The main treatment approach is focused on infection control (Ravikumar et al., 2021). Additionally, there is a lack of clear diagnostic methods and prognostic indicators for SC (L’Heureux et al., 2020). Therefore, the discovery of new biomarkers and the investigation of the molecular mechanisms involved in myocardial cell death in SC are of great significance for further research and clinical management of this condition.

Pyroptosis is a unique proinflammatory programmed cell death mode that causes cell membrane perforation through gasdermin family proteins, leading to cell swelling and death (Yu et al., 2021). At the same time, pyroptotic cells can release inflammatory factors such as IL-1β and IL-18, triggering an inflammatory response. Pyroptosis is associated with the pathogenesis of various human diseases, including cardiovascular diseases. Research in the field of cardiovascular diseases is rapidly expanding, highlighting the role of pyroptosis in the development and progression of these diseases. One study demonstrated that GSDMD-mediated pyroptosis promotes myocardial cell damage during ischemia-reperfusion (I/R) (Shi et al., 2021), and another suggested that NLRP3-mediated pyroptosis is involved in the pathogenesis of nonischemic dilated cardiomyopathy (Zeng et al., 2020). These findings indicate that the process of pyroptosis formation is closely linked to cardiovascular diseases. Recently, an increasing number of studies have investigated the relationship between pyroptosis and sepsis (Liu et al., 2022; Kang et al., 2018; Zheng et al., 2021), suggesting that pyroptosis may underlie myocardial injury in SC, and pyroptosis-related genes (PRGs) may be the targets for the diagnosis, prognosis, and treatment of SC.

The objective of this study is to investigate PRGs in SC and explore the mechanism of pyroptosis involved in the regulation of SC formation and progression. To this end, the SC-related dataset GSE79962 was downloaded from the Gene Expression Omnibus (GEO) database and analyzed using bioinformatics methods. The expression pattern of PRGs in SC was further identified based on their original gene expression profiling data. miRNAs, transcription factors (TFs), and therapeutic drugs targeting these PRGs were also predicted. Moreover, the expression of the identified PRGs was verified by the external datasets and the LPS-induced cell damage models. Additionally, the relationship between the PRGs and pyroptosis in the LPS-induced cell damage model was explored by knockdown experiments.

Materials & Methods

Data source

The SC-related microarray datasets GSE79962 (mRNA), GSE53007 (mRNA), and GSE142615 (mRNA) were obtained from the Gene Expression Omnibus (GEO) database (https://www.ncbi.nlm.nih.gov/). GSE79962 is based on the GPL6244 platform ([HuGene-1_0-st] Affymetrix Human Gene 1.0 ST Array [transcript (gene) version]). GSE53007 is based on the GPL6885 platform (Illumina MouseRef-8 v2.0 expression beadchip). GSE142615 is based on the GPL27951 platform (Agilent-084783 Mus musculus array (circMouse_0410; Agilent Probe Name)). The samples in GSE79962 were derived from human heart tissue, comprising twenty samples from patients with septic cardiomyopathy and eleven samples from non-heart failure donors. In GSE53007, the samples were obtained from mouse heart tissue, including four sepsis mice and four healthy mice. As for GSE142615, the samples were collected from heart tissue of mice, consisting of four LPS-induced septic cardiomyopathy mice and four saline-treated control mice.

Differential expression analysis

GEO2R is an interactive online tool for analyzing various gene expression datasets (https://www.ncbi.nlm.nih.gov/geo/geo2r/). Based on the cut-off criteria: P < 0.05 & |logFC| > 0.5, GEO2R was utilized to identify differentially expressed genes (DEGs). The volcano plot of these DEGs and the heat map based on the top 50 DEGs with them respective P-values were obtained via the online bioinformatics analysis platform Sanger Box 3.0 (http://www.sangerbox.com).

Functional and pathway enrichment analysis

Gene Ontology (2015) (GO) analysis and Kyoto Encyclopedia of Genes and Genomes (Kanehisa et al., 2017) (KEGG) pathway enrichment analysis of the identified DEGs were conducted using the R package “clusterProfiler” (version 4.4.4) in R software (version 4.2.1; R Core Team, 2022). Pathways with P < 0.05 were considered to be significant.

Expression pattern of the differentially expressed PRGs in SC

The 108 PRGs, including drivers, suppressors, and markers, were obtained from previous studies (Song et al., 2021; Zeng et al., 2021) (File S1). The DEGs obtained from GSE79962 were intersected with these PRGs to obtain the differentially expressed PRGs. Violin plot was used to illustrate the differences in PRGs expression between groups in GSE79962. Furthermore, a correlation heatmap was generated to demonstrate the correlation among the differentially expressed PRGs in GSE79962. GO annotation and KEGG pathway enrichment analysis were performed to determine the biological function of the differentially expressed PRGs. Additionally, the Reactome database was utilized for further pathway enrichment analysis of the differentially expressed PRGs.

Related biological process of differentially expressed PRGs

Gene Set Enrichment Analysis (GSEA) (Subramanian et al., 2005) was performed on candidate diagnostic biomarkers to better understand their biological processes. Based on the expression levels of candidate diagnostic biomarkers, the samples were divided into high-expression and low-expression groups. To evaluate the relevant pathways and molecular mechanisms, gene sets of “c2.cp.kegg.v7.4.symbols.gmt” were downloaded from MSigDB (http://www.gsea-msigdb.org/gsea/downloads.jsp).

Prediction of PRG-related miRNAs, transcription factors, and targeted therapeutic drugs

Online databases, including TargetScan (https://www.targetscan.org/vert_80/), miRDB (https://mirdb.org/), miRWalk (http://mirwalk.umm.uni-heidelberg.de/), and TarBase v.3 (https://dianalab.e-ce.uth.gr/html/diana/web/index.php?r=tarbasev8) were utilized to predict miRNAs targeting the differentially expressed PRGs. The prediction was based on the miRNAs predicted by at least three of the four databases. In addition, the TFs of the differentially expressed PRGs were predicted using TRRUST (https://www.grnpedia.org/trrust/) and RegNetwork (https://regnetworkweb.org/) databases, where the TFs predicted by both databases were considered as the TFs of the PRGs. The CTD database (http://ctdbase.org/) was employed for predicting therapeutic drugs targeting differentially expressed PRGs. Finally, Cytoscape software was used to visualize the predicted results in interaction networks.

Further validation of differentially expressed PRGs in external datasets

The GSE53007 and GSE142615 datasets contained transcriptomic data from myocardial tissues of septic cardiomyopathy mice and control mice. The expression of PRGs in SC was studied using raw expression data from GSE53007 and GSE142615 datasets. Statistical analysis was conducted using independent sample t-tests, and P < 0.05 was considered statistically significant.

Cell culture and treatment

The rat cardiomyocyte H9C2 cell line was obtained from BeNa culture collection (Beijing, China) and cultured in complete medium containing Dulbecco’s modified Eagle’s medium (DMEM; Gibco, Billings, MT, USA) with 10% fetal bovine serum (FBS; Gibco, Billings, MT, USA) and 1% penicillin-streptomycin (Sigma-Aldrich, St. Louis, MO, USA) at 37 °C with 5% CO2. Lipopolysaccharide (LPS; Sigma-Aldrich) was dissolved in sterile deionized water. The H9C2 cardiomyocytes were cultured in DMEM with LPS 1 µg/ml, 2.5 µg/ml, 5 µg/ml, 7.5 µg/ml, 10 µg/ml for 24 h to find the optimal concentration (10 mg/ml) for inducing the myocardial cell damage model in vitro. The H9C2 cardiomyocytes of the control group were cultured in the complete medium containing an equal volume of sterile deionized water for 24 h. The human cardiomyocyte AC16 cell line was also obtained from BeNa culture collection (Beijing, China) and cultured in complete medium at 37 °C with 5% CO2. AC16 cell line was then cultured in complete medium containing a concentration of 10 µg/ml LPS for 24 h to establish an LPS-induced damage model of AC16 cells. The AC16 cardiomyocytes of the control group were also cultured in the medium containing an equal volume of sterile deionized water for 24 h.

Cell viability assay

Cell viability was determined by Cell Count Kit-8 assay (CCK-8; Beyotime, China). Cells were seeded in 96-well plates at a density of 1 × 104 cells per well in 100 µl complete medium. For H9C2 cells, different gradient concentrations of LPS (1, 2.5, 5, 7.5, 10 µg/ml) were added, while AC16 cells were exposed to 10 mg/ml LPS. The cells were then incubated for 24 h. After that, 10 µl CCK-8 solution was added to each well and incubated for another 2 h. Finally, the optical density (OD) value of each well was measured at 450nm using the microplate reader (PerkinElmer, Turku, Finland). Cell activity histograms were drawn according to OD value. Statistical analysis was conducted using independent sample t-tests, and P < 0.05 was considered statistically significant.

Hoechst 33342 and propidium iodide (PI) staining

To detect cell pyroptosis (Shi et al., 2021; He et al., 2020; Lv et al., 2023), Hoechst 33342/PI staining was used on LPS-induced cells and control cells. The cells were seeded in six-well plates at a density of 2 × 105 cells per well and cultured for 24 h of incubation. Then, the cells were separately exposed to complete medium containing 10 µg/mL concentration of LPS and an equal volume of sterile deionized water for 24 h. Subsequently, they were washed twice with pre-cooled PBS. The mixture of Hoechst 33342/PI (1 ml staining buffer + 5 µl Hoechst 33342 + 5 µl PI) was incubated with the cells in the dark at 4 °C for 20 min. The stained cells were immediately captured using an inverted fluorescence microscope (Olympus, Tokyo, Japan). The PI stained cells were quantitatively analyzed by Image J (Version 1.53e) software. Three independent technical replicates were performed in each group. Statistical analysis was conducted using independent sample t-tests, and P < 0.05 was considered statistically significant.

siRNA knockdown of SOD2 in AC16 cells

The AC16 cells were seeded in six-well plates at a density of 2 × 105 cells per well. After 24 h, siRNA transfection was performed. Small interfering RNA (siRNA) targeting human SOD2 (5′-GCACGCUUACUACCUUCAGUAdTdT-3′) (Genecreate, Wuhan, China) and negative control (5′-UUCUCCGAACGUGUCACGUdTdT-3′) (Genecreate, Wuhan, China) were diluted to 50nM. To transfect the siRNA into the cells, 2 µL of Lipofectamine 2000 (Gibco, Billings, MT, USA) was mixed with 200 µL Opti-MEM (Gibco, Billings, MT, USA) and left to incubate at room temperature for 5 min. Next, 3 µL of siRNA was mixed with 800 uL Opti-MEM. The mixture of Lipofectamine 2000 and Opti-MEM was combined with the mixture of siRNA and Opti-MEM and incubated at room temperature for 20 min. Afterwards, the mixture was added to the cells. After incubating the cells for 6 h at 37 °C with 5% CO2, the medium was replaced with DMEM containing 10% FBS. The cells were further incubated at 37 °C with 5% CO2 for 24 h to complete the siRNA knockdown.

Real-time quantitative polymerase chain reaction

The expression of genes in this study was detected at the mRNA level using real-time quantitative polymerase chain reaction (qPCR). Total RNA in the cells was extracted using the FastPure Cell/Tissue Total RNA Isolation Kit V2 (Vazyme, Nanjing, China). Afterwards, the mRNA was reverse transcribed into cDNA using the Hifair III 1st Strand cDNA Synthesis SuperMix for qPCR (Ye Sen, Shanghai, China) in a PCR amplifier (Bio-Rad, Hercules, CA, USA). The qPCR was performed using Hieff UNICON Universal Blue qPCR SYBR Green Master Mix (Ye Sen, China) on the Bio-Rad CFX96 Real-time PCR Detection System, and then ΔCt values of the genes were obtained. β-actin was used as the reference gene, and the relative fold change of genes was calculated using the 2−ΔΔCt method. File S2 includes the detailed information of all primers. Data were presented as mean values with standard error of the mean (SEM) from at least three independent experiments. Statistical analysis was conducted using independent sample t-tests, and P < 0.05 was considered statistically significant.

Results

Identification of DEGs and key pathways in SC

The SC-related dataset GSE79962 was obtained from the Gene Expression Omnibus (GEO) database. Based on the screening critieria P < 0.05 and |logFC| > 0.5, 1206 DEGs were screened, with 663 genes showing high expression and 543 genes showing low expression (File S3). Figure 1 presents the volcano plot of DEGs (Fig. 1A) and the heatmap displaying the top 50 DEGs based on the P values (Fig. 1B). Using the R package “clusterProfiler”, KEGG pathway enrichment analysis and GO annotation were performed to characterize the biological function of these DEGs. Figures 1C–1F presents the top fifteen enriched KEGG pathways and GO annotation terms. “Non-alcoholic fatty liver disease (NAFLD)”, “Mineral absorption”, “HIF-1 signaling pathway”, “Ferroptosis”, and “Malaria” were the important pathways in KEGG enrichment analysis (Fig. 1C). The Biological Process (BP) category of the GO annotation results showed that these DEGs were significantly enriched in the terms of “response to organic substance”, “small molecule metabolic process”, “positive regulation of multicellular organismal process”, “cell activation”, “circulatory system development” (Fig. 1D). For the Cellular Component (CC) category of the GO annotation, the top five significantly enriched terms were “extracellular vesicle”, “extracellular organelle”, “mitochondrion”, “mitochondrial part”, and “organelle envelope” (Fig. 1E). The top five significantly enriched terms of GO Molecular Function (MF) included “catalytic activity”, “signaling receptor binding”, “protein dimerization activity”, “transporter activity”, and “protein homodimerization activity” (Fig. 1F).

Figure 1 (A–F) Identification of DEGs and key pathways in SC.

Expression pattern of the differentially expressed PRGs in SC

Ten differentially expressed PRGs (SOD2, GJA1, TIMP3, TAP1, TIMP1, NOD1, TP53, CPTP, CASP4, SAT1) were identified by intersecting DEGs with PRGs obtained from previous studies. The Venn diagram displaying the intersection is shown in Fig. 2A. The violin plot illustrates the expression differences of these PRGs between SC and control groups (Fig. 2B). The PRGs highly expressed in the SC group were SOD2, TAP1, TIMP1, NOD1, TP53, CASP4, and SAT1, while low expression in the SC group was observed for GJA1, TIMP3, and CPTP. Correlation analysis revealed that in PRGs, SOD2 had the strongest positive correlation with SAT1 (r = 0.84), while GJA1 and NOD1, GJA1 and TP53, SAT1 and CPTP all had the strongest negative correlation (r =  − 0.59) (Fig. 2C). GO annotation showed that these PRGs were mainly involved in “metabolic process regulation”, “cell death”, “programmed cell death”, and “apoptosis process” (Fig. 2D). KEGG pathway enrichment analysis revealed that these PRGs were primarily associated with “Ferroptosis”, “Longevity regulating pathway”, and “NOD-like receptor signaling pathway” (Fig. 2E). Additionally, reactome pathway analysis demonstrated that the “Pyroptosis” pathway exhibited the highest enrichment of PRGs (Fig. 2F).

Figure 2 (A–F) Expression pattern of differentially expressed PRGs.

Related biological process of candidate diagnostic biomarkers

To gain a deeper understanding of the pathways associated with the identified PRGs, GSEA analysis was conducted. The top five GSEA results based on KEGG pathways for each of the ten PRGs are shown in Figs. 3A–3J. The most relevant pathways associated with each PRGs are as follows: NOD1 was associated with bladder cancer (Enrichment Score (ES) = −0.6213, Nominal P value (NP) = 0.0000), GJA1 with cardiac muscle contraction (ES = 0.4938, NP = 0.0078), TP53 with acute myeloid leukemia (ES = −0.5429, NP = 0.002), CASP4 with apoptosis (ES = −0.5017, NP = 0.0000), SOD2 with bladder cancer (ES = −0.5906, NP = 0.0000), TAP1 with cell cycle (ES = −0.4438, NP = 0.0039), TIMP1 with pathway in cancer (ES = −0.4174, NP = 0.0000), TIMP3 with glycosphingolipid biosynthesis globo series (ES = −0.6076, NP = 0.0020), SAT1 with amino sugar and nucleotide sugar metabolism (ES = −0.6724, NP = 0.0000), and CPTP with alzheimers disease (ES = 0.4170, NP = 0.0225).

Figure 3 (A–J) Related biological process of differentially expressed PRGs.

Prediction of PRG-related miRNAs, transcription factors, and targeted therapeutic drugs

miRNAs, TFs, and therapeutic drugs targeting these PRGs were predicted using relevant databases. The results showed that a total of 147 miRNAs targeted these PRGs, and a miRNA-mRNA regulatory network with 157 nodes and 152 edges was constructed (Fig. 4A). Among the predicted miRNAs, hsa-miR-21-3p, hsa-miR-30d-3p, hsa-miR-4492, hsa-miR-6829-5p, and hsa-miR-4685-5p targeted more than two PRGs. Additonally, a total of 31 TFs were found to target these PRGs, and a TF-mRNA network with 36 nodes and 38 edges was constructed (Fig. 4B). Among the predicted TFs, RELA, STAT1, NFKB1, SP1, and JUN targeted more than two PRGs. Moreover, thirteen drugs targeting more than eight PRGs were selected, such as Resveratrol, Cisplatin, Dexamethasone and Gentamicins. A drug-mRNA network with 23 nodes and 113 edges was constructed (Fig. 4C).

Figure 4 (A–C) Prediction of PRG-related miRNAs, transcription factors, and targeted therapeutic drugs.

Further validation of differentially expressed PRGs in external datasets GSE53007 and GSE142615

The ten differentially expressed PRGs (SOD2, GJA1, TIMP3, TAP1, TIMP1, NOD1, TP53, CPTP, CASP4, SAT1) were further evaluated through the use of external datasets (mouse). In GSE53007, NOD1, SOD2, TIMP3, TAP1, TIMP1, CASP4, and SAT1 were highly expressed in the SC group, GJA1 was low expressed in the SC group, and there was no significant difference in TP53 between the SC and the control group (except for CPTP, which could not be investigated through the annotation platform) (Fig. 5A). In GSE142615, NOD1, SOD2, TIMP3, TAP1, TIMP1, CASP4, and SAT1 were highly expressed in LPS-induced group, GJA1 expression was low among LPS-induced group. There were no significant differences on TP53 and CPTP between the LPS-induced group and the control group (Fig. 5B). The expression trends of NOD1, SOD2, GJA1, TAP1, TIMP1, CASP4 and SAT1 in these two external datasets were consistent with the original analysis results.

Figure 5 (A–B) Further validation of differentially expressed PRGs in external datasets.

Cell viability and pyroptosis detection in H9C2 cells

The cell viability of H9C2 cells treated with various concentrations of LPS was assessed using CCK-8. The results showed that the viability of H9C2 cells decreased with the increase of LPS concentration, and the maximum reduction was observed when treated with 10 µg/mL LPS (Fig. 6A). Subsequently, Hoechst 33342/PI was employed to stain 10 µg/mL LPS-induced cells and control cells to identify pyroptotic cells with incomplete cell membranes. The results demonstrated a significant increase in the number of pyroptotic cells with incomplete cell membranes in the LPS-induced group compared to the control group (Figs. 6B–6C).

Figure 6 (A–C) Cell viability, cell pyroptosis staining in the LPS-induced H9C2 cell damage model.

qRT-PCR verification of PRGs in LPS-induced H9C2 and AC16 myocardial cell damage model

The expression of the ten differentially expressed PRGs (SOD2, GJA1, TIMP3, TAP1, TIMP1, NOD1, TP53, CPTP, CASP4, SAT1) in the LPS-induced H9C2 (rat) myocardial cell damage model were detected by qPCR. The results showed that four PRGs: SOD2 (up), TIMP1 (up), TIMP3 (up), and TP53 (up) had significant differences in expression between the LPS-induced group and control group (Fig. 7A). According to the intersection of qPCR results from H9C2 cells and external dataset validation, SOD2 and TIMP1 were confirmed to be consistent with the predicted results (Fig. 7B). Finally, further validation of SOD2 and TIMP1 was conducted in the LPS-induced AC16 cell damage model using qPCR. The results indicated that SOD2 expression is up-regulated in LPS-induced AC16 cells (Fig. 7C).

Figure 7 (A–C) qRT-PCR verification of PRGs in LPS-induced H9C2 and AC16 cells.

Cell viability and pyroptosis detection in SOD2 knocked-down AC16 cells

To investigate the impact of SOD2 knockdown on pyroptosis in SC, LPS-induced cell damage model was established using AC16 cells with SOD2 knockdown and negative control (NC). qPCR confirmed the successful knockdown of SOD2 in AC16 cells by siRNA (Fig. 8A). Cell viability was assessed using CCK-8, and the results revealed a significant reduction in cell viability in the SOD2 knockdown group compared to the NC group under LPS-stimulated conditions (Fig. 8B). Although there was no difference in cell viability between the NC group and the LPS-stimulated NC group in AC16 cells, the expression of inflammatory markers IL6 and TNF α was significantly increased in the LPS-stimulated NC group (Figs. 8C–8D). This indicated the successful establishment of the LPS-induced AC16 cell damage model. Pyroptotic cells were detected using Hoechst 33342/PI staining, and the results indicated that SOD2 knockdown increased pyroptosis in the LPS-induced AC16 cell damage model (Figs. 8E–8F).

Figure 8 (A–F) Cell viability and pyroptosis detection in the LPS-induced cell damage model of SOD2-knockdown AC16 cell.

Discussion

Sepsis, a life-threatening organ dysfunction caused by a faulty host response to infection, remains a significant contributor to critically ill patients worldwide. Among the complications arising from sepsis, septic cardiomyopathy stands out as a notable disorder of heart function. However, due to the complexities of sepsis-induced cardiovascular symptoms, there is still no clear definition and diagnostic criteria for septic cardiomyopathy (Hollenberg & Singer, 2021). Consequently, it becomes imperative to delve deeper into the pathogenesis of septic cardiomyopathy and identify reliable biomarkers of myocardial injury during sepsis.

Pyroptosis is a form of programmed cell death mediated by gasdermin, characterized by cell swelling and the release of inflammatory cytokines. It has been established that microbial infections such as viruses and bacteria can induce pyroptosis through the canonical pathway mediated by inflammasome (Bergsbaken, Fink & Cookson, 2009). At present, Pyroptosis is thought to be involved in the progression of various cardiovascular diseases (Zhaolin et al., 2019), such as atherosclerosis (Xu et al., 2018), ischemic heart disease (Kawaguchi et al., 2011), diabetic cardiomyopathy (Luo et al., 2017), and cardiac hypertrophy (Bai et al., 2018). Despite this progress, the precise mechanism of pyroptosis in SC is still not fully elucidated. That is why aiming at exploring the PRGs’ patterns of expression and molecular mechanisms in SC, we conducted this public databases SC-related datasets bioinformatics analysis.

In this study, we conducted bioinformatics analyses on SC-related datasets to delve into the expression patterns and molecular mechanisms of the PRGs in SC. A total of 1,206 DEGs were screened from the human SC-related dataset GSE79962. GO annotation and KEGG pathway enrichment analysis showed that these DEGs were mainly enriched in the pathways of “metabolism”, “energy generation”, “oxidation–reduction process”, “circulatory system development”, “mitochondria”, “inflammatory response”, “Mineral absorption”, “HIF-1 signaling pathway” and “Ferroptosis”. The results of enrichment analysis indicate that metabolism and energy generation are closely related to the pathogenesis of SC. A large number of studies have shown that in the development of sepsis, the response of innate immune cells to systemic inflammation leads to oxidative stress, resulting in the generation of ROS and mitochondrial dysfunction (Arulkumaran et al., 2016; Mantzarlis, Tsolaki & Zakynthinos, 2017; Almalki, 2021). Secondary to shock hypoperfusion and mitochondrial destruction, the body would depend mainly on anaerobic glycolysis for energy production. This would promote lactic acid-related metabolism disorders (Garcia-Alvarez, Marik & Bellomo, 2014). Sepsis cardiomyopathy is also closely related to metabolism and mitochondrial function. It has been reported that increased glycolysis and mitochondrial dysfunction could induce apoptosis, thus, leading to septic cardiomyopathy (Zheng et al., 2017). Ferroptosis has been shown to be involved in disease progression in septic cardiomyopathy. Ferroptosis was observed in LPS-stimulated septic mice as well as in H9C2 cardiomyocytes (Li et al., 2020), and inhibition of ferroptosis by Ferrostatin-1 and dexmedetomidine significantly decreased cardiac damage in septic mice (Wang et al., 2020a; Wang et al., 2020b; Xiao et al., 2021).

Pyroptosis is involved in septic-related cardiac insufficiency. Stimulator of interferon gene (STING) activates NLRP3 in an interferon regulatory factor 3 (IRF3) dependent manner in LPS-treated mice, inducing pyroptosis via the canonical pathway (Li et al., 2019a; Li et al., 2019b). Zinc finger antisense 1 (ZFAS1), which is activated by the transcription factor SP1, can induce pyroptosis in cardiomyocytes by targeting the miR-590-3p/AMPK/mTOR signal, and aggravate cardiac dysfunction caused by sepsis (Liu et al., 2020). Furthermore, pyroptosis is involved in other organ damage caused by sepsis. Caspy2 can cause pyroptosis by activating GSDMEb, and the caspy2-GSDMEb pathway plays a critical role in LPS-induced renal tubule injury in zebrafish sepsis (Wang et al., 2020a; Wang et al., 2020b). In a CLP-induced acute pulmonary edema model, inhibiting HMGB1 expression reduces caspase11-dependent pyroptosis in lung tissue, thereby reducing lung injury (Xie et al., 2021).

In GSE79962, ten differentially expressed PRGs were screened. Enrichment analysis showed that these PRGs were involved in “cell death”, “programmed cell death”, “apoptotic process”, and “Ferroptosis”. The ten PRGs were further validated by external datasets. The result demonstrated that the expressions of NOD1, SOD2, TIMP3, TAP1, TIMP1, CASP4, and SAT1 were significantly increased in the SC group compared to the control group.

Many studies have reported on the role of these PRGs in pyroptosis. GJA1 (CX43) is a gap connexin involved in cell growth and apoptosis. It was demonstrated that GJA1 knockdown significantly increased the expression of active caspase-1 in X-ray-induced human umbilical vein endothelial cells and promotes pyroptosis phenomenon (Li et al., 2019a; Li et al., 2019b). CASP4 is a key gene in the non-canonical pathway of pyroptosis. CASP4 is activated by the LPS present in the cells, cleaving GSDMD into D-GSDMD with some perforation characteristics and leads to the formation of cell membrane pores (Aglietti et al., 2016; Shi et al., 2014). TP53-induced glycolysis and apoptosis regulator (TIGAR) can inhibit microglial pyroptosis in hypoxic-ischemic brain damage (Tan et al., 2021). A study has shown that SOD2 can inhibit ROS production, thereby alleviating ROS-induced pyroptosis in non-small cell lung cancer (Liu et al., 2019). However, it is worth mentioning that the correlation between these PRGs and pyroptosis in SC still needs to be confirmed.

In this study, we generated an in vitro model of cell injury by treating myocardial cells with LPS to simulate the state of septic cardiomyopathy. We then conducted preliminary analysis of pyroptosis using Hoechst 33342/PI staining and observed a significant increase in the number of pyroptotic cells in the LPS-induced model. We validated the expression of these PRGs using both SC-related external datasets and qPCR in LPS-induced models. Finally, our results proved that SOD2 was up-regulated in SC. SOD2 is a crucial antioxidant enzyme primarily present in the inner mitochondrial membrane, and reactive oxygen species (ROS) is a pivotal factor in the activation of the NLRP3 inflammasome. Previous studies have established the involvement of the SOD2-ROS axis in pyroptosis in various disorders (Jiang et al., 2021; Yang et al., 2023; Dong et al., 2021). During oxidative stress, the increased expression of SOD2 enhances the clearance of ROS, thereby reducing NLRP3 activation and, to some extent, alleviating pyroptosis. In our investigation, we observed that mitochondrial energy production dysfunction is associated with the advancement of septic cardiomyopathy. Next, we observed that SOD2 knockdown in human cardiomyocytes led to a significant increase in pyroptosis upon LPS stimulation compared to normal cardiomyocytes stimulated with LPS. This indicates that the SOD2-ROS axis can regulate pyroptosis in cardiomyocytes. Hence, there is a high likelihood that SOD2-mediated pyroptosis may play a role in the molecular mechanism of septic cardiomyopathy.

This study has some restrictions because it was only based on published datasets on transcriptomes related to SC and cell experiments. In future, animal models can be built to further study the mechanism of pyroptosis in SC, and blood samples from patients can be collected to confirm the diagnostic utility of these PRGs.

Conclusions

A total of 1,206 DEGs in heart tissue samples from SC patients were screened and these genes are mainly involved in metabolism and mitochondrial function. We identified ten PRGs and predicted 147 miRNAs, 31 TFs and 13 therapeutic drugs targeting these PRGs. SOD2 was confirmed to be a potential regulatory target of pyroptosis in SC. Knockdown of SOD2 resulted in increased pyroptosis in the LPS-induced AC16 cell damage model.

Supplemental Information

File S1 Pyroptosis-related genes obtained from previous studies

Click here for additional data file.

File S2 The primer sequence information of qPCR experiment

Click here for additional data file.

File S3 Information of the DEGs in GSE79962

Click here for additional data file.

File S4 The raw data of qPCR

Click here for additional data file.

We sincerely appreciate the researchers for providing their GEO database information online, we are truly honored to acknowledge their contributions. We are also grateful to the Sanger Box online biomedical data analysis tool for simplifying the analysis process.

List of Abbreviations

BP Biological Process

CC Cellular Component

DEG Differentially expressed genes

GEO Gene Expression Omnibus

GO Gene Ontology

GSEA Gene Set Enrichment Analysis

KEGG Kyoto Encyclopedia of Genes and Genomes

MF Molecular Function

PRG Pyroptosis-Related genes

SC Septic Cardiomyopathy

TF Transcription Factors

Additional Information and Declarations

Competing Interests

Author Contributions

Data Availability

The authors declare there are no competing interests.

Haoyan Zhu conceived and designed the experiments, performed the experiments, analyzed the data, prepared figures and/or tables, and approved the final draft.

Jiahe Wu conceived and designed the experiments, performed the experiments, analyzed the data, prepared figures and/or tables, and approved the final draft.

Chenze Li analyzed the data, authored or reviewed drafts of the article, and approved the final draft.

Ziyue Zeng analyzed the data, authored or reviewed drafts of the article, and approved the final draft.

Tianwen He analyzed the data, authored or reviewed drafts of the article, and approved the final draft.

Xin Liu analyzed the data, authored or reviewed drafts of the article, and approved the final draft.

Qiongxin Wang analyzed the data, authored or reviewed drafts of the article, and approved the final draft.

Xiaorong Hu conceived and designed the experiments, authored or reviewed drafts of the article, and approved the final draft.

Zhibing Lu conceived and designed the experiments, authored or reviewed drafts of the article, and approved the final draft.

Huanhuan Cai conceived and designed the experiments, authored or reviewed drafts of the article, and approved the final draft.

The following information was supplied regarding data availability:

The data is available at NCBI GEO: GSE79962, GSE53007, GSE142615.

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
