# Peer review of "Transcriptome analysis reveals the mechanism of pyroptosis-related genes in septic cardiomyopathy"

_PeerJ, doi:10.7717/peerj.16214_

## Round 0.1 · original submission · Major Revisions

(1) The sub-figures need to be re-arranged. It may not be enough to improve their high-resolution only. The best criterion is probably that we can read the words in the figures clearly in 100% view of the PDF document on A4.

(2) P to indicate the statistical significance need to be italicized.

(3) A space is always required between the values and units.

(4) Some references are missing page numbers or article numbers (For example, Bai Y. et al. 2018, BIOSCIENCE REPORTS 38). In addition, please unify the format of all references (such as the journal name). Please carefully check all the references before resubmission.

·

Basic reporting

1) The figures provided in the manuscript are difficult to read due to low resolution.

2) Some abbreviations used in the manuscript require the inclusion of their full names when they are first mentioned. For instance, line 48, “PRG”; line 54, “ssGSEA”; line 67, “LPS”; line 107, “ssGSEA”; line 161, “ssGSEA”, the full name should be moved to the previous part; line 242 “BP”; line 245, “CC”; line 248, “MF”; line 272, “ES” and “NP”.

Experimental design

1) Missing information about the version of R packages and software used. Please update the manuscript and add all this information.

Validity of the findings

no comment

Additional comments

Tentative statement: moderate revision

In this work, the authors aim to identify the biomarker genes for septic cardiomyopathy through the implementation of various tools, such as DE analysis, GSEA, ssGSEA, and online databases. Overall, the manuscript is well-written. However, there are a few issues more that need to be addressed.

1) Line 129, “Based on the cut-off criteria: P < 0.05 …”. Is the P here the original p-value or adjusted p-value? Please make it clear. If the original p-value is used, why don’t the author use the adjusted p-value since multiple tests were implemented here.
2) Line 193, “… was cultured in …” change to “… and was cultured in …”.

·

Basic reporting

The manuscript was not professionally written.The manuscript has several grammatical and format errors that could impact its quality. It's advisable for the author to engage a professional editing service to eliminate these errors and enhance the presentation of the manuscript.

Experimental design

This manuscript employed various bioinformatic tools to investigate pyroptosis-related genes in septic cardiomyopathy. The analysis was conducted using published datasets specifically related to septic cardiomyopathy. The research's scope and objective were initially well-defined; however, it is recommended to narrow the focus. The inclusion of "Identification of the Immune Infiltration Landscape in SC" does not align with the intended purpose of this study.

Validity of the findings

The study results addressed the research question, but more emphasis should be placed on the focus. However, relying solely on another dataset for the validation of differentially expressed PRGs was not convincing. It would have been preferable to experimentally explore the function of these PRGs in septic cardiomyopathy, such as through overexpression or knockdown in cardiac cells.

Additional comments

Minor concerns should be addressed.
1. It seems there may be a mistake in the legend of Fig1B. The correct data set should be GSE79962 instead of GSE26155.
2. In Fig1A and B, should label the groups information (Nonfailing donor heart or Septic cardiomyopathy heart).
3. In Fig1C, the genes should be labeled according to the KEGG pathway.
4. Fig4 is unrelated to the purpose of this study, as previously mentioned. It is better to delete this part.
5. In Fig6, in addition to conducting bioinformatic analyses on other data sets, it would be advantageous to experimentally investigate the function of these PRGs in septic cardiomyopathy. This could involve performing overexpression or knockdown experiments in cardiac cells, providing a more comprehensive understanding of their role in the disease.
6. In Fig7,It is better to use human cardiac cell line rather than rat H9C2 cell line.

Reviewer 3 ·

Basic reporting

Good reporting format.

Experimental design

1. Please clarify whether you used adjusted p-values in the gene expression analysis using GEO2R. If adjusted p-values were not used, please provide a justification for this choice in the context of your study.

2. It would be beneficial to include an output table in the supplementary material that lists the identified differentially expressed genes (DEGs) along with their corresponding p-values. This would provide readers with a comprehensive view of the results.

3. Please specify the correlation method used in your study. Providing this information will enhance the transparency and reproducibility of your findings.

Validity of the findings

The study aims to identify pyroptosis-related genes (PRGs) in SC and explore their involvement in SC formation and progression. The study validates SOD2 and TIMP1 as potential targets for diagnosing and treating SC using relatively solid evidence. Overall, the study presents valuable insights into the mechanisms underlying septic cardiomyopathy.

---

## Round 0.2 · accepted · Accept

A couple of suggestions for proofreading:

(1) Reviewer 1 suggested that the resolution of figures can be further improved. I agreed with his point of view. At least Figure 1 should be further updated to be more clearer.

(2) The number (108) of pyroptosis-related genes shown in the text (line 143) is not consistent with that (122) in Supplementary File 1. Please correct it.

·

Basic reporting

no comment

Experimental design

no comment

Validity of the findings

no comment

Additional comments

The authors have addressed most of my comments. The resolution of figures can be further improved.

·

Basic reporting

no comment

Experimental design

no comment

Validity of the findings

no comment

Additional comments

I recognize and value the authors' sincere efforts in shaping this manuscript. They have effectively addressed the previous concerns.

Reviewer 3 ·

Basic reporting

The revised manuscript is clear and unambiguous. The revised paper provided sufficient field background and context, and literature references. The paper has a good format including nice figures and tables. Also, the paper is self-contained with relevant results to hypothesis.

Experimental design

After the revision, the experimental design is good. Specifically, the research questions were well defined which is relevant and meaningful. In this paper, rigorous investigation was done to show a high technical and ethical standard. Also, the methods in this paper were described with sufficient details and information to replicate.

Validity of the findings

Since the methods and results parts were revised based on the reviewers' comments, the results and conclusions were relatively promising. In the revised paper, the underlying data have been provided and the data are robust. The conclusions are well stated and linked to the research questions.

Additional comments

I recommend to accept the paper.